# Effects of miR-306 Perturbation on Life Parameters in the English Grain Aphid, *Sitobion avenae* (Homoptera: Aphididae)

**DOI:** 10.3390/ijms25115680

**Published:** 2024-05-23

**Authors:** Linyuan Wu, Guohua Wei, Yi Yan, Xuguo Zhou, Xun Zhu, Yunhui Zhang, Xiangrui Li

**Affiliations:** 1State Key Laboratory for Biology of Plant Diseases and Insect Pests, Institute of Plant Protection, Chinese Academy of Agricultural Sciences, Beijing 100193, China; wulinyuan0525@163.com (L.W.); w2950390082@163.com (G.W.); y3352853039@163.com (Y.Y.); zhuxun@caas.cn (X.Z.); 2Department of Entomology, School of Integrative Biology, College of Liberal Arts & Sciences, University of Illinois Urbana-Champaign, Urbana, IL 61801, USA; xgzhou@illinois.edu

**Keywords:** *Sitobion avenae*, miR-306, growth and development, two-sex life table, population prediction

## Abstract

MicroRNAs (miRNA) play a vital role in insects’ growth and development and have significant potential value in pest control. Previously, we identified miR-306 from small RNA libraries within the English grain aphid, *Sitobion avenae*, a devasting insect pest for wheat. miR-306 not only involves in wing morphogenesis, but also is critically important for aphid survival. Its specific impacts on the life history traits, however, remain unclear. Here, we evaluate the impact of miR-306 perturbation on *S*. *avenae* populations using a two-sex life table approach. This comprehensive analysis revealed that miR-306 perturbation significantly prolongs the developmental stages (9.64% and 8.20%) and adult longevity of *S*. *avenae*, while decreasing pre-adult survival rate (41.45% and 38.74%) and slightly reducing average fecundity (5.80% and 13.05%). Overall, miR-306 perturbation negatively affects the life table parameters of the aphid population. The population prediction models show a significant decline in the aphid population 60 days post interference, compared to the control groups (98.14% and 97.76%). Our findings highlight the detrimental effects of miR-306 perturbation on *S*. *avenae* population growth and suggest potential candidate genes for the development of RNAi-based biopesticides targeted specifically at this pest species.

## 1. Introduction

The English grain aphid (*Sitobion avenae*) is a significant global pest of wheat, causing negative impacts on grain crop production, and resulting in substantial economic losses [1,2]. Both nymph and adult grain aphids feed on phloem fluids from wheat plants during the wheat seedling and jointing stages, exacerbating nutrient deficiencies, and consequently decreasing grain yields. Additionally, they secrete honeydew, which covers the leaf surface, and impedes plant respiration and photosynthesis. This ultimately results in reduced wheat quality and yield losses of up to 30–40% [3,4,5]. As a typical *r*-selected pest, *S. avenae* has a short lifecycle and rapid population growth. Furthermore, its long-distance migratory behavior allows the rapid expansion of infestations, thus compounding the challenges of pest management. Currently, the prevention and control of grain aphids predominantly rely on chemical pesticides [6,7]. Frequent use of these pesticides, however, promotes resistance development, increasing the risk of pest outbreaks [6]. This underscores the urgent need for innovative and sustainable control strategies.

RNA interference (RNAi), first discovered in the nematode *Caenorhabditis elegans*, involves the sequence-specific suppression of gene expression in response to double-stranded RNA (dsRNA) [8,9]. Regarded as a revolutionary advance in pest control, RNAi technology facilitates the development of sustainable integrated pest management (IPM) strategies [9,10,11,12,13]. Compared to traditional chemical pesticides, RNAi-based biopesticides offer numerous advantages, including high target specificity, environmental degradability, diverse targeting options, and design flexibility [14]. For example, the trend of mortality was significantly higher in *Spodoptera frugiperda* larvae pretreated with dsRNA (*CYP302A1*) followed by the exposure of rice plants at 72 and 96 h [15]. The apterous adult aphids became more sensitive to clothianidin, with a higher mortality rate than the control after silencing *CYP6CY14* and *CYP6DC1* genes [16]. Furthermore, in order to improve RNAi efficiency, researchers have developed innovative nanomaterial carriers, such as the star polycation (SPc). This efficient and cost-effective nanocarrier not only protects dsRNA but also significantly enhances RNAi efficacy [17].

MicroRNAs (miRNAs) are a class of endogenous, single-stranded RNA molecules approximately 22 nucleotides in length, prevalent across various organisms and playing pivotal roles in gene expression regulation. Most miRNAs modulate gene expression by interacting with the untranslated regions (UTR) or open reading frames (ORF) of target genes with their seed regions, through either complete or incomplete complementarity. This interaction influences the transcriptional regulation of genes, affecting insect gene functionality [18,19,20,21,22]. miRNA inhibitors, such as mimic (agomir) or inhibitor (antagomir), can be introduced into insects through feeding, injection, or drip infusion, offering novel approaches for pest control [23,24,25]. For instance, miR-2703 targets *CHS1a* for chitin synthesis regulation in response to the 20E signaling pathway in brown planthopper, with the injecting miR-2703 mimic reducing the chitin synthesis, increasing mortality, and causing molting defects [26]. Similarly, miR-184 regulates the transcription level of *CYP303A1* during the molting process in locusts, leading to molting abnormalities and death [27]. In pea aphid, *Acyrthosiphon pisum*, overexpression of miR-92a-1-p5 decreases *flightin* gene expression, resulting in wing deformities and reducing flight capabilities [28]. In bird cherry-oat aphid, *Rhopalosiphum padi*, miR-147b targets and downregulates the *vg* gene expression, reducing the proportion of winged individuals [29]. A versatile microRNA, miR-306, influences the development of insect wings and halteres, and the terminal differentiation of male germline in *Drosophila* [30,31,32]. When grain aphids *S. avenae* are fed with miR-306 agomir, it results in wing development abnormalities and increased mortality [33]. These studies offer a promising strategy to control aphid populations from impacting their mobility and flight ability, by interfering with the expression of relevant miRNAs.

To comprehensively assess the effects of miR-306 agomir on the population dynamics of *S. avenae*, we employed the nanomaterial SPc/adjuvant carrier to precisely deliver miR-306 agomir and interfere to newly hatched nymphs within 24 h. Using the two-sex life table method, which has provided valuable insights into the population ecology of target species, serving as essential tools for assessing population fitness and dynamics [25,26,27,28,29], we investigated the impact of miR-306 agomir on the growth, development, and reproductive capability of *S. avenae*. This research aims to contribute to the development of miRNA-based nucleic acid pesticides and provide new molecular targets and methodologies for effective aphid control.

## 2. Results

### 2.1. Developmental Duration of S. avenae

Different treatments led to variations in the lifecycles of *S. avenae*. The entire nymphal stage for *S. avenae* treated with miR-306 perturbation was significantly extended (9.02 d) compared to those treated with the water control group NCW (8.15 d, *p* < 0.00001) and the nanomaterial control group NCS (8.28 d, *p* = 0.00002). Specifically, the developmental duration for the 3rd and 4th instar nymphs under miR-306 perturbation (2.20 d and 2.41 d, respectively) was significantly longer than those under NCW (1.99 d, 2.13 d, *p* = 0.01929 and 0.00043) and NCS (1.97 d, 2.17 d, *p* = 0.01361 and 0.00612) treatments (Table 1). The developmental durations for other nymphal instars did not differ significantly across treatments. In the adult stage, the longevity of *S. avenae* treated with miR-306 perturbation (14.26 d) was significantly longer than that observed in NCW (11.64 d, *p* = 0.00038) and NCS (12.25 d, *p* = 0.00397). Contrarily, the mean longevity of *S. avenae* treated with miR-306 perturbation was significantly shorter than both NCW (*p* = 0.00001) and NCS (*p* = 0.00003) treatments. The pre-adult survival rate for *S. avenae* treated with miR-306 was 52.69%, which was significantly lower than that for NCW (89.99%, *p* < 0.00001) and NCS (86.01%, *p* < 0.00001), respectively. Additionally, significant differences were observed in the total pre-oviposition period (TPOP) for the miR-306 perturbation group. Average fecundity in the miR-306 treated group (23.73 individuals) was significantly lower than in the NCS group (27.29 individuals, *p* = 0.04532) but not significantly different from the NCW group (25.19 individuals, *p* = 0.40729). Furthermore, the proportion of winged morphs under miR-306 perturbation decreased significantly (1.33%) compared to those under NCW (13.33%, *p* = 0.00071) and NCS (6.67%, *p* = 0.03002) treatments.

### 2.2. Life Table Parameters

Life table parameters for *S. avenae* under different treatments revealed significant differences (Table 2). The intrinsic rate of increase (0.1728 d^−1^) for the miR-306-treated group was significantly lower than NCW (0.2375 d^−1^, *p* < 0.00001) and NCS (0.2351 d^−1^, *p* < 0.00001). Similarly, the finite rate of increase (1.1886 d^−1^) was significantly lower compared to NCW (1.2681 d^−1^, *p* < 0.00001) and NCS (1.2650 d^−1^, *p* < 0.00001). The net reproductive rate (12.50 d^−1^) was significantly lower than NCW (22.67 d^−1^, *p* < 0.00001) and NCS (23.48 d^−1^, *p* < 0.00001). Additionally, the mean generation time (14.60 d) was significantly longer than that for NCW (13.14 d, *p* < 0.00001) and NCS (13.42 d, *p* < 0.00001).

### 2.3. Survival Curves of S. avenae under Different Treatment Conditions

The age-stage specific survival rate (*S_xj_*) curve of the *S. avenae* population represents the probability of survival from the 1st instar to age *x* and stage *j* (Figure 1). Among the three treatment conditions, the survival rate of *S. avenae* was highest in the NCW group, followed by NCS. The lowest survival rate was observed in the group treated with miR-306 perturbation. Despite identical rearing conditions, the overlapping nature of the *Sxj* curves across different stages indicate variability in the developmental rate among individuals within the population.

The age-specific survival rate curve (*l_x_*, Figure 2) for *S. avenae* under different treatment conditions provides a simplified overview of the *S_xj_* curve, focusing solely on survival rate across all instars at each age *x*. It is clear that *S. avenae* treated with miR-306 perturbation experienced a significant decline in the age-specific survival rate curve between days 3 and 7, indicating a higher mortality rate among the younger instars. Although differences in survival times among treatment conditions were not statistically significant, miR-306 perturbation-treated *S. avenae* adults all died by day 33, while the control groups NCW and NCS experienced complete mortality at 33 d and 31 d, respectively (Figure 2).

### 2.4. Reproductive Rate Curves of S. avenae under Different Treatment Conditions

The reproductive dynamics of female aphids are depicted through the age-stage reproductive rate curve (*f_xj_*), age-specific reproductive rate curve (*m_x_*), and age-specific net fecundity curve (*l_x_m_x_*), all of which demonstrate consistent trend (Figure 3). The *f_xj_* curve indicates the average number of offspring produced by all female adults on day *x*, with the peak point indicating the highest daily aphid production. The average daily aphid production reached its maximum on the 11th, 14th, and 12th days for miR-306 perturbation, NCS, and NCW treatments, respectively, with values of 2.70, 3.29, and 2.74 offspring per day. The peak points for mx were 2.59, 3.29, and 2.72, while for *l_x_m_x_*, they were 1.37, 2.59, and 2.34, respectively.

### 2.5. Simulation of S. avenae Population Projection

Population projection for *S. avenae* under three different treatments was simulated using TIMING-MSchart software (Version 2023.06.26) (Figure 4). After 20 d, the logarithmic scale growth curve of the *S. avenae* population approached linearity, indicating a stable age-stage distribution. The linear population growth was represented by the slope of the regression line, corresponding to lg*λ*. Among the treatments, the population growth rate for *S. avenae* treated with miR-306 perturbation was the slowest, with the predicted population size of approximately 0.11 million individuals at 60 d, representing a reduction of 98.14% compared to the NCW (5.92 million), and 97.76% compared to the NCS (4.9 million).

## 3. Discussion

RNAi is emerging as a promising genetic approach for pest management [11,34], capable of suppressing genes that are crucial for the physiological functions of insects, potentially leading to their lethality [35,36,37,38]. RNAi offers an advantage over previous transgenic methods by significantly broadening the range of target genes for pest control [39]. The expression of pest miRNA can be effectively disrupted through RNAi, which can be utilized to control pests [23,24]. Here, we employ a two-sex life table method to demonstrate that miR-306 perturbation can cause high mortality rates in *S*. *avenae* populations.

Additionally, the results of this study reveal that all three treatments allowed the *S. avenae* population to complete a single generation cycle. The application of miR-306 perturbation notably extended the nymphal stage duration, TPOP, while significantly reducing the pre-adult survival rate. These effects are similar to those observed with the use of conventional insecticides like imidacloprid on grain aphids and acetamiprid on cotton aphids [40,41,42]. Thus, the miR-306 perturbation, akin to traditional chemical pesticides, adversely affects the growth, development, survival, and reproduction of *S. avenae*, highlighting its potential for practical application in aphid control within agricultural settings.

Furthermore, the study found that the net reproductive rate (*R*_0_), finite rate of increase (*λ*), and intrinsic rate of increase (*r*) of *S. avenae* treated with miR-306 perturbation were lower compared to controls. Additionally, the mean generation time (*T*) was longer in the treated groups. The lower net reproductive rate can be attributed to a higher mortality rate among nymphs during the larval stage (Table 1 and Figure 1), while the extended mean generation time results from prolonged nymphal and adult stages in miR-306-treated *S. avenae*. These patterns have been corroborated by previous studies. For instance, soybean aphids exposed to a sublethal concentration of 0.2 mg/L of imidacloprid exhibited reductions in *R*_0_, *r*, and *λ*, compared to the control groups [43]. Moreover, sulfoxaflor was found to decrease same reproductive parameters in cotton aphids and significantly extend the average generation time [44]. Chlorantraniliprole treatment also reduced these parameters in *Plutella xylostella*, significantly prolonging the mean generation time [45]. These observations suggest that the miR-306 perturbation disrupts population dynamics in a manner comparable to that of conventional chemical pesticides, adversely affecting the life history traits of *S. avenae* and impeding population growth. Population projection results further confirm that miR-306 perturbation is detrimental to the growth of the *S. avenae* population.

The stage frequency curves (*S_xj_*; Figure 1) provide a detailed overview of the survival and stage differentiation of aphids under virous conditions. While it was challenging to quantify the exact decline at each stage in our study, the simplified version of the age-stage survival rate (*Sxj*; Figure 1) [46] revealed a noticeable reduction in the specific survival rates of *S. avenae* treated with miR-306 perturbation. Specifically, there was a significant decline in survival rates in the week prior to treatment (*l_x_*; Figure 2), highlighting the impact of miR-306 perturbation on aphid survival. Similar effects were observed by Chen [44], where cotton aphids treated with a sulfoxaflor showed a rapid decrease in specific age survival rates (*l*_x_) on the 3rd day. Additionally, the *m_x_* results from the miR-306 perturbation-treated group suggest that pest managers should prioritize interventions during the peak reproductive period to mitigate increased pest pressure. These findings demonstrate a significant inhibitory effect of miR-306 perturbation on the development population dynamics of the *S. avenae*.

miR-306 plays a crucial regulatory role in various biological processes. In the oncological research on fruit flies, miR-306 was found to inhibit tumor growth and induce apoptosis by activating the JNK signaling pathway, suggesting its potential as a therapeutic target [32]. Moreover, the regulatory relationship between miR-306 and the abrupt gene in insect development has been elucidated. By modulating abrupt expression, miR-306 plays a critical role in the development of insect wings and halteres, underlining its importance in morphogenesis and offering new avenues for further research into insect development [31]. Additionally, miR-306-5p has been shown to regulate chitin metabolism in *Aedes albopictus* pupae through the linc8338-miR-306-5p-XM_019678125.2 axis, providing insights into insect exoskeleton formation [47]. In our study, miR-306 not only induced mortality and prolonged the developmental period of the grain aphid, but also reduced the winged population rate and inhibited population fertility, suggesting that miR-306 could be a viable target for developing biopesticides aimed at controlling grain aphids. Given miR-306’s diverse functions across different organisms, further research may uncover additional regulatory mechanisms and targets, offering new strategies for disease treatment and advancements in biological research.

Two-sex-specific life tables are an effective method for evaluating the impacts of environmental factors on insect population dynamics [48]. Historically applied across various ecological and physiological studies, life tables analyze population responses under varying conditions such as constant and fluctuating temperatures [30,31], different host plant impacts [32,33,34,35], and genetic factors like inbreeding [33]. Recently, the use of bisexual life tables has expanded, becoming crucial in studies of insect physiology and biochemistry, particularly for integrated pest management research. In this study, the detrimental effects of miR-306 perturbation on the life parameters of a laboratory population of *S. avenae* were clearly demonstrated using the life table method. However, additional studies are necessary to explore the cumulative effects on subsequent generations. It is important to note that this research focused solely on laboratory population, while field populations and external environmental factors such as temperature, light, and precipitation could significantly influence the efficacy of the miR-306 perturbation. Consequently, a more comprehensive assessment is required to fully understand the influence of miR-306 perturbation on natural populations of *S. avenae*.

## 4. Materials and Methods

### 4.1. Laboratory Aphid Population

*S. avenae* were collected from fields of common wheat, *T. aestivum*, in Langfang, Hebei Province, China (39°3′42″ N, 16°3′67″ E), during the 2012 growing season. Aphids were transported to the laboratory and reared on 15 cm tall wheat seedlings under controlled conditions of 20 °C, 60%RH, and 16L:8D photoperiod. To ensure genetic consistency, a healthy aphid from the rearing cages was selected for clonal rearing across three generations. The fourth generation (G4) was utilized for experimental trials.

### 4.2. Chemical Agents

miR-306 agomir and NC agomir (Appendix A) [33] were synthesized by Shanghai Gima Pharmaceutical Technology Co., Ltd. (Shanghai, China) Nanomaterials SPc and alkyl glycosides (adjuvant) were provided by Professor Shen Jie’s team at China Agricultural University; double distilled water (ddH_2_O) was supplied by Beijing Tiangen Biochemical Technology has Limited company (Beijing, China); 90 mm plastic petri dishes from Guangzhou Shuopu Biotechnology Co., Ltd. (Guangzhou, China); qualitative filter paper from Hangzhou Special Paper Co., Ltd. Company (Hangzhou, China); and microdispenser from Hamilton, Switzerland.

### 4.3. Experimental Design

To determine the optimal dose of miR-306 agomir for *S. avenae* interfering using a nanomaterial delivery system, a preliminary experiment was conducted. First-instar nymphs were treated with 0.2 μL of miR-306 agomir at various concentrations (200–1000 nmol/L). Mortality rates peaking at 400 nmol/L and stabilized thereafter. Consequently, 0.2 μL of 400 nmol/L was chosen as the optimal dose. To prepare the treatment agents, mix 1 OD_260_ (approximately 33 μg) of miR-306 agomir or NC agomir dry powder with 125 μL ddH_2_O to obtain an original solution of 20 μM. Dilute 5 μL of this solution to 400 nmol/L, then add an equivalent mass of nanomaterial SPc and 0.5% of the total volume as an additive.

The experiment was divided into three treatment groups: the miR-306 group: miR-306 agomir (400 nmol/L) + ddH_2_O + nanomaterial + adjuvant; the water control group (NCW): ddH_2_O; and the nanomaterial control group (NCS): NC agomir (400 nmol/L) + nanomaterial + adjuvant. Each group included 150 newborn 1st instar aphids, with each receiving 0.2 μL of the respective treatment.

The miR-306 perturbation was applied to the dorsal side of the aphids. After the inhibitor was fully absorbed, aphids were carefully transferred to a 90 mm plastic petri dish containing a fresh wheat seedling (2~4 cm in length) using a paintbrush. The seedlings were sprayed with water regularly to maintain moisture and replaced every 3 days. Daily observations were made to monitor molting and maturation, offspring were counted daily, and the nymphs were removed until the adults perished.

### 4.4. Data Analysis

The population parameters were analyzed using the following equations:

Net reproductive rate (*R*_0_):(1)$$R_{0}=\sum_{x=0}^{\infty }l_{x}m_{x}$$

Intrinsic rate of increase (*r*), using Euler–Lotka equation with the iterative bisection method, starting from the birth of the aphid [49]:(2)$$\sum_{x=0}^{\infty }e^{-r(x+1)}l_{x}m_{x}$$

Finite rate of increase (*λ*):(3)$$\lambda =e^{r}$$

Mean generation time (*T*):(4)$$T=\frac{\ln(R_{0})}{r}$$

The mean values and standard errors of the population parameters, including the mean longevity of 1st to 4th instar nymphs and adults, the adult and total pre-reproductive period, and mean fecundity, were analyzed using the TWOSEX-MSChart software (Version 2023.05.07) [50]. To assess statistical differences among treatment groups, a paired bootstrap test (B = 100,000) was employed [51,52,53,54]. This test evaluates the differences based on the percentile and the 95% confidence interval (CI) of the normalized distribution of differences [55,56]. To project the total population growth, the initial *S. avenae* population, comprising 150 newborn nymphs, was analyzed over a period of 60 d using the TIMING-MSChart program (Version 2023.06.26) [50]. This approach allows for the prediction of population dynamics based on the parameters derived from the experimental data. All graphical representations of curves were generated using GraphPad Prism 8.3, ensuring high-quality visualizations of the data.

## 5. Conclusions

This study utilized RNA interference (RNAi) technology within the theoretical and methodological framework of sex-specific life tables to analyze the impact of miR-306 perturbation on the life parameters of *S. avenae* in a controlled laboratory environment. The interference of miR-306 extended the entire developmental period of the *S. avenae* population, reduced pre-adult survival and reproductive capacity, significantly decreased the net reproductive rate (*R*_0_), intrinsic rate of increase (*r*), and finite rate of increase (*λ*), and significantly prolonged the average generation time (*T*). These findings unequivocally demonstrate that miR-306 perturbation has a detrimental effect on the growth, development, survival, and reproduction of *S. avenae*. Furthermore, the simulation of population dynamics over 60 days, based on derived population parameters, revealed that the population treated with miR-306 perturbation exhibited the smallest population size. This confirms the strong inhibitory effect of miR-306 perturbation on the population growth of *S. avenae*. In conclusion, this study not only identifies miR-306 as a potential target for the development of RNAi-based biopesticides to control *S. avenae*, but it also underscores the broader implications for the management and control of this pest species. This research provides valuable insights into the molecular mechanisms by which RNAi can be harnessed to disrupt critical life processes in agricultural pests, offering a strategic approach to pest management.

## Figures and Tables

**Figure 1 ijms-25-05680-f001:**
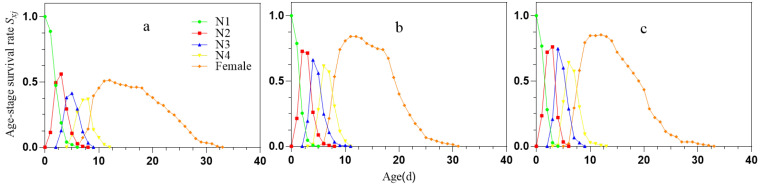
Age-stage specific survival rate curves (*Sxj*) of *S. avenae*. (**a**) miR-306 agomir treatment; (**b**) NCS treatment; (**c**) NCW treatment: N1: 1st instar nymph; N2: 2nd instar nymph; N3: 3rd instar nymph; N4: 4th instar nymph; Female: Female adult.

**Figure 2 ijms-25-05680-f002:**
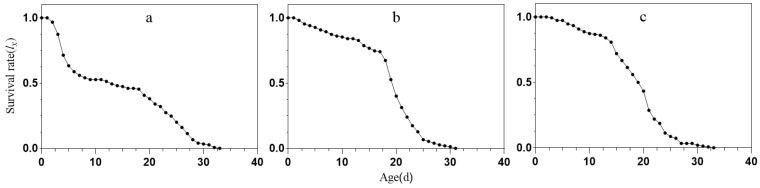
Age-specific survival rate curves (*l_x_*) of *S. avenae*. (**a**) miR-306 agomir agent treatment; (**b**) NC agomir agent treatment; (**c**) Water treatment.

**Figure 3 ijms-25-05680-f003:**
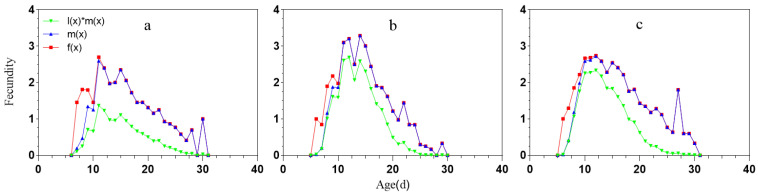
Female age-stage-specific fecundities (*f_xj_*), age-specific fecundities (*m_x_*) and age-specific net maternities (*l_x_m_x_*) of *S. avenae*. (**a**) miR-306 agomir agent treatment; (**b**) NC agomir agent treatment; (**c**) Water treatment.

**Figure 4 ijms-25-05680-f004:**
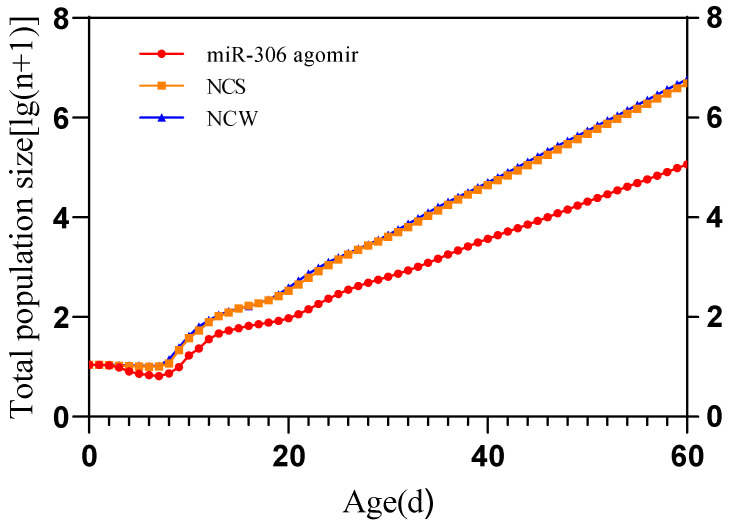
Population projections of different stages of *S. avenae* [lg(n + 1)].

**Table 1 ijms-25-05680-t001:** Mean values (±SE) of the developmental duration of *S. avenae*.

Life Parameters	NCW	NCS	miR-306
	Mean ± SE	Mean ± SE	Mean ± SE
1st instar nymph (N1) (d)	2.07 ± 0.06 a	2.06 ± 0.07 a	2.16 ± 0.07 a
2nd instar nymph (N2) (d)	1.98 ± 0.04 a	2.07 ± 0.06 a	2.13 ± 0.08 a
3rd instar nymph (N3) (d)	1.99 ± 0.05 b	1.97 ± 0.06 b	2.20 ± 0.07 a
4th instar nymph (N4) (d)	2.13 ± 0.05 b	2.17 ± 0.06 b	2.41 ± 0.06 a
Pre-adult (d)	8.15 ± 0.10 b	8.28 ± 0.09 b	9.02 ± 0.14 a
Adult longevity (d)	11.64 ± 0.41 b	12.25 ± 0.33 b	14.26 ± 0.61 a
Mean longevity (d)	18.46 ± 0.49 a	18.41 ± 0.53 a	14.40 ± 0.83 b
Pre-adult survival (%)	89.99 ± 2.45 a	86.01 ± 2.84 a	52.69 ± 4.09 b
TPOP (d)	8.64 ± 0.12 b	8.91 ± 0.12 b	9.60 ± 0.16 a
APOP (d)	0.50 ± 0.05 a	0.63 ± 0.06 a	0.57 ± 0.07 a
Mean fecundity (individuals)	25.19 ± 1.14 ab	27.29 ± 1.10 a	23.73 ± 1.39 b

Note: miR-306, miR-306 agomir (400 nmol/L) + nanomaterial + adjuvant; NCW, ddH_2_O; NCS, NC agomir (400 nmol/L) + nanomaterial + adjuvant; APOP, adult pre-reproductive period; TPOP, total pre-reproductive period. Mean ± SE (standard error) followed by different letters in the same row are significantly different when calculated using the paired bootstrap test at the *p* < 0.05 level.

**Table 2 ijms-25-05680-t002:** Life table parameters of *S. avenae* treated with miR-306 perturbation.

Parameters	NCW	NCS	miR-306
	Mean ± SE	Mean ± SE	Mean ± SE
Intrinsic rate of increase (*r*) (d^−1^)	0.2375 ± 0.0041 a	0.2351 ± 0.0047 a	0.1728 ± 0.0077 b
Finite rate of increase (*λ*) (d^−1^)	1.2681 ± 0.0052 a	1.2650 ± 0.0059 a	1.1886 ± 0.0091 b
Net reproductive rate (*R*_0_)	22.67 ± 1.45 a	23.48 ± 1.22 a	12.50 ± 1.22 b
Mean generation time (*T*) (d)	13.14 ± 0.15 b	13.42 ± 0.14 b	14.60 ± 0.21 a

Note: miR-306, miR-306 agomir (400 nmol/L) + nanomaterial + adjuvant; NCW, ddH_2_O; NCS, NC agomir (400 nmol/L) + nanomaterial + adjuvant; APOP, adult pre-reproductive period; TPOP, total pre-reproductive period. Mean ± SE (standard error) followed by different letters in the same row are significantly different when calculated using the paired bootstrap test at the *p* < 0.05 level.

## Data Availability

The data are available upon request.

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
