# Peer review of "Effects of miR-306 Perturbation on Life Parameters in the English Grain Aphid, *Sitobion avenae* (Homoptera: Aphididae)"

_ijms, 2024, doi:10.3390/ijms25115680_

Round 1
Reviewer 1 Report
Comments and Suggestions for Authors
Comments for author
In this work, the authors proposed an interesting study “Effects of miR-306 interference on life parameters in the English grain aphid, Sitobion avenae (Homoptera: Aphididae)”. The proposal is so interesting evaluate the impact of miR-306 interference on S. avenae populations using a two-sex life table approach. This comprehensive analysis revealed that miR-306 interference significantly prolongs the developmental stages and adult longevity of S. avenae, while decreasing pre-adult survival rate and a slightly reducing average fecundity. Overall, miR-306 interference negatively affects the life table parameters of the aphid population. The population prediction models show a significant decline in the aphid population days of post-interference. Our findings highlight the detrimental effects of miR-306 interference on S. avenae populations growth and suggest potential candidate genes for the development of RNAibased biopesticides targeted specifically at this pest species. Some details must be revised, mainly on references, but the work is supported by the results and the proper literature. In this sense, the manuscript can be after minor revisions.
General Comments:
The manuscript is well-written, technically accurate, and adheres to the expected article structure. The literature references and background are appropriate, providing valuable insights into the molecular mechanisms by which RNAi can disrupt critical life processes in agricultural pests, thus offering a strategic approach to pest management.
Experimental Design:
The background adequately supports the research question, and the experimental design is appropriate.
Validity of the Findings:
The results are relevant and obtained with accuracy according to the expected experimental design, thereby supporting the conclusions. The findings of this research offer valuable insights into the molecular mechanisms by which RNAi can disrupt critical life processes in agricultural pests, providing a strategic approach to pest management.
Minor Considerations:
1. Line: Please ensure that important quantitative values are included in the abstract.
2. Please revise the keywords.
3. Please correct the paragraph format in the introduction section and throughout the manuscript. Additionally, please add more references to the introduction, such as PMID: 36762173, PMID: 36127062.
4. Lines 98-99: Please clarify the meaning of this sentence.
5. How did the author calculate the concentrations of miR-306 agomir?
6. Were preliminary assays performed?
7. Which insect stage did the author use for the experiment?
8. Please add minor ticks to the Y-axis of all figures.
9. Updated references should be included in the discussion. Many studies in similar research areas have been reported, such as DOI: 10.1127/entomologia/2023/2002.
10. The quality of sentences should be consistent and professional throughout the manuscript. Please ensure that grammar and language are thoroughly checked for errors.
Author Response
RE: Manuscript ID ijms-3005960
Manuscript ijms-3005960 entitled “Effects of miR-306 interference on life parameters in the English grain aphid, Sitobion avenae (Homoptera: Aphididae)” has been revised according to the editorial suggestions. Reviewers’ constructive criticisms and suggestions are greatly appreciated. We have essentially followed most of the editorial changes suggested by the reviewers, and the remaining comments and suggestions have been incorporated into the revised manuscript (see attached Final Revision for details). The following is a point-to-point response to reviewers’ comments:
Reviewer(s)’ Comments to Author:
Reviewer #1
In this work, the authors proposed an interesting study “Effects of miR-306 interference on life parameters in the English grain aphid, Sitobion avenae (Homoptera: Aphididae)”. The proposal is so interesting evaluate the impact of miR-306 interference on S. avenae populations using a two-sex life table approach. This comprehensive analysis revealed that miR-306 interference significantly prolongs the developmental stages and adult longevity of S. avenae, while decreasing pre-adult survival rate and a slightly reducing average fecundity. Overall, miR-306 interference negatively affects the life table parameters of the aphid population. The population prediction models show a significant decline in the aphid population days of post-interference. Our findings highlight the detrimental effects of miR-306 interference on S. avenae populations growth and suggest potential candidate genes for the development of RNAi-based biopesticides targeted specifically at this pest species. Some details must be revised, mainly on references, but the work is supported by the results and the proper literature. In this sense, the manuscript can be after minor revisions.
RESPONSE: We appreciate reviewer’s comments.
-Line: Please ensure that important quantitative values are included in the abstract.
RESPONSE: Done.
-Please revise the keywords.
RESPONSE: Done.
-Please correct the paragraph format in the introduction section and throughout the manuscript. Additionally, please add more references to the introduction, such as PMID: 36762173, PMID: 36127062.
RESPONSE: We have corrected the paragraph format in the introduction section and throughout the manuscript. Additionally, we incorporated the recommended references PMID: 36762173 [15] and PMID: 36127062 [16]. [2] [3] [6] [7] [9] and [14] into the introduction, and have highlighted them in yellow on the reference list.
-Lines 98-99: Please clarify the meaning of this sentence.
RESPONSE: Revisions have been made following reviewer’s suggestions on page 3, line 103-105.
-How did the author calculate the concentrations of miR-306 agomir? Were preliminary assays performed?
RESPONSE: Regarding the calculation of miR-306 agomir concentrations, we initially mixed 1 OD260 (equivalent to approximately 33 μg) of miR-306 agomir dry powder with 125 μL ddH2O to obtain a stock solution of 20 μM. Subsequently, for experimental purposes, we diluted 5 μL of this stock solution to achieve a working concentration of 400 nmol/L, as detailed on page 7, lines 274-278 of our manuscript. Prior to commencing our main study, we conducted preliminary experiments to validate the accuracy of our dilution procedure and ensure the reliability of the resulting concentrations.
-Which insect stage did the author use for the experiment?
RESPONSE: The first instar nymphs (N1) was used. This information has been provided on page 7, line 282 of our manuscript
-Please add minor ticks to the Y-axis of all figures.
RESPONSE: Revisions have been made following reviewer’s suggestions.
-Updated references should be included in the discussion. Many studies in similar research areas have been reported, such as DOI: 10.1127/entomologia/2023/2002.
RESPONSE: we incorporated the recommended references DOI: 10.1127/entomologia/2023/2002 [25] [37] [38] and [39] into the discussion, and have highlighted them in yellow on the reference list.
-The quality of sentences should be consistent and professional throughout the manuscript. Please ensure that grammar and language are thoroughly checked for errors.
RESPONSE: Done.
Reviewer 2 Report
Comments and Suggestions for Authors
The manuscript by Wu et al describes impacts on population dynamics following delivery of a mimic of miR-306 to aphids. In one of their previous studies, they identified this microRNA, and noted that feeding miR-306 agomirs caused abnormal wing development and increased mortality. In this study, they further examined impacts on survival and fecundity of topical delivery of the agomirs and used life table analyses to assess the potential to use these microRNA mimics to control aphid populations. The study was well described, the paper was clearly written, and the authors have fairly noted the value of life table evaluations to assess the impacts of the agomirs. I have only several questions that the authors should address before publication:
1. Throughout the manuscript, the impacts of the agomir are defined as interference of miR-306. While interference can be used broadly, the word is more often used to describe knockdown of a transcript by RNAi. Given that agomirs are agonists, I think it would be more appropriate to define the changes in miR-306 as “perturbations”, not “interference”.
2. The mode of action of the agonist has not yet been confirmed in this study. i.e. how does the addition of the agonist cause death or reduced fecundity in the aphids? In your previous study, increased mortality was already noted, so the decline of the population in this study is not surprising. But how might this miR-306 affect fecundity? Using qRT-PCR to assess that the delivered agomir was having the intended effect on some target genes would be highly informative.
3. The sequence of the agomir has not been defined in the Methods.
4. How was the agomir complexed with Star-polycation nanoparticles? The ratio of miRNA:nanoparticle is not described.
5. Lines 283-284. How was molting assessed? Was there a method to detect increased size or were exuviae counted?
6. How was the duration of each instar calculated? From the Methods, 150 aphids were used in each treatment, but how did you determine when each one molted if they were all together? A single daily observation may not provide you with the precision you need to discriminate between small differences in molting times.
Comments on the Quality of English LanguageThere are only some typos that need correction.
Author Response
RE: Manuscript ID ijms-3005960
Manuscript ijms-3005960 entitled “Effects of miR-306 interference on life parameters in the English grain aphid, Sitobion avenae (Homoptera: Aphididae)” has been revised according to the editorial suggestions. Reviewers’ constructive criticisms and suggestions are greatly appreciated. We have essentially followed most of the editorial changes suggested by the reviewers, and the remaining comments and suggestions have been incorporated into the revised manuscript (see attached Final Revision for details). The following is a point-to-point response to reviewers’ comments:
Reviewer(s)’ Comments to Author:
Reviewer #2
The manuscript by Wu et al describes impacts on population dynamics following delivery of a mimic of miR-306 to aphids. In one of their previous studies, they identified this microRNA, and noted that feeding miR-306 agomirs caused abnormal wing development and increased mortality. In this study, they further examined impacts on survival and fecundity of topical delivery of the agomirs and used life table analyses to assess the potential to use these microRNA mimics to control aphid populations. The study was well described, the paper was clearly written, and the authors have fairly noted the value of life table evaluations to assess the impacts of the agomirs.
RESPONSE: We appreciate reviewer’s comments.
-Throughout the manuscript, the impacts of the agomir are defined as interference of miR-306. While interference can be used broadly, the word is more often used to describe knockdown of a transcript by RNAi. Given that agomirs are agonists, I think it would be more appropriate to define the changes in miR-306 as “perturbations”, not “interference”.
RESPONSE: Done.
-The mode of action of the agonist has not yet been confirmed in this study. i.e. how does the addition of the agonist cause death or reduced fecundity in the aphids? In your previous study, increased mortality was already noted, so the decline of the population in this study is not surprising. But how might this miR-306 affect fecundity? Using qRT-PCR to assess that the delivered agomir was having the intended effect on some target genes would be highly informative.
RESPONSE: We understand reviewer’s concerns. In our current study, we have focused primarily on examining the effects of miR-306 agomir on survival rate, growth /development, reproduction, and population dynamics in aphids. However, we recognize the importance of elucidating the molecular mechanisms underlying miR-306 agomir-induced mortality and reduced fecundity in aphids, which we intend to pursue in the subsequent experiments. Our ultimate goal is to understand the pest control potential of miR-306 in insects in general and aphids in particular.
-The sequence of the agomir has not been defined in the Methods.
RESPONSE: We have included the sequence information for the miR-306 agomir and NC agomir in Table S1.
Table S1. Sequence information of miRNA agomir
|
Name |
Sequences (5’-3’) |
|
miRNA-306 agomir |
UCAGGUACCAAGUGAUUUCUGA AGAAAUCACUUGGUACCUGAUU |
|
NC agomir |
UUCUCCGAACGUGUCACGUTT ACGUGACACGUUCGGAGAATT |
-How was the agomir complexed with Star-polycation nanoparticles? The ratio of miRNA:nanoparticle is not described.
RESPONSE: Agomir and Star cationic nanoparticles are compounded in a 1:1 mass ratio, as detailed on page 7, line 277.
-Lines 283-284. How was molting assessed? Was there a method to detect increased size or were exuviae counted?
RESPONSE: In this study, the aphids were reared individually in 90mm culture dishes. We assessed their developmental stages and age changes primarily by counting the number of molts, as each molt signifies a transition to a new instar. After recording the number of molts, we took care to promptly remove the exuvia (molted skin) to avoid confusion and ensure accurate tracking of the aphids' developmental progress.
-How was the duration of each instar calculated? From the Methods, 150 aphids were used in each treatment, but how did you determine when each one molted if they were all together? A single daily observation may not provide you with the precision you need to discriminate between small differences in molting times.
RESPONSE: To calculate the duration of each instar, aphids were reared individually in 90 mm diameter culture dishes. We observed each aphid daily to document the time of molting, as molting signifies the transition to the next developmental stage.